# COVID-19 incidence and outcome by affluence/deprivation across three pandemic waves in Ireland: A retrospective cohort study using routinely collected data

**Declan McKeown**[ID][1]☯*, **Angela McCourt**[1]☯, **Louise Hendrick**[1]‡, **Anne O'Farrell**[1]☯, **Fionnuala Donohue**[1]‡, **Laurin Grabowsky**[1]☯¤, **Paul Kavanagh**[1]‡, **Patricia Garvey**[2]☯, **Joan O'Donnell**[2]☯, **Lois O'Connor**[2]‡, **John Cuddihy**[2]‡, **Matt Robinson**[1]‡, **Declan O'Reilly**[1]‡, **Anthony Staines**[ID][3]‡, **Howard Johnson**[1]☯

**1** National Health Intelligence Unit, Health Service Executive, Dublin, Ireland, **2** Health Protection Surveillance Centre, Health Service Executive, Dublin, Ireland, **3** School of Nursing, Psychotherapy and Community Health, Dublin City University, Dublin, Ireland

☯ These authors contributed equally to this work.
¤ Current address: Currently with Health Protection Surveillance Centre, Health Service Executive, Dublin, Ireland
‡ LH, FD, PK, LO, JC, MR, DO, and AS also contributed equally to this work.
* declan.mckeown@hse.ie

**Data Availability Statement:** Data cannot be shared publicly because of General Data Protection

## Abstract

### Background

Since the pandemic onset, deprivation has been seen as a significant determinant of COVID-19 incidence and mortality. This study explores outcomes of COVID-19 in the context of material deprivation across three pandemic waves in Ireland.

### Methods

Between 1st March 2020 and 13th May 2021, 252,637 PCR-confirmed COVID-19 cases were notified in Ireland. Cases were notified to the national Computerised Infectious Disease Reporting (CIDR) system. Each case was geo-referenced and assigned a deprivation category according to the Haase-Pratschke (HP) Deprivation Index. Regression modelling examined three outcomes: admission to hospital; admission to an intensive care unit (ICU) and death.

### Results

Deprivation increased the likelihood of contracting COVID-19 in all age groups and across all pandemic waves, except for the 20–39 age group. Deprivation, age, comorbidity and male gender carried increased risk of hospital admission. Deprivation was not a factor in predicting ICU admission or death, and diagnosis in wave 2 was associated with the lowest risk of all three outcomes.

Regulations (GDPR) which have been introduced in Europe since 2018. Because the Health Protection Surveillance Centre are the designated Data Controllers under GDPR, any request for data would need to be made through them. Data requests can be sent to the director of HPSC, Dr. Greg Martin gregory.martin@hpsc.ie, or gregory.martin@hse.ie

**Funding:** The authors received no specific funding for this work.

**Competing interests:** The authors have declared that no competing interests exist.

## Conclusions

Our study suggests that COVID-19 spreads easily through all strata of society and particularly in the more deprived population; however this was not a consistent finding. Ireland is ethnically more homogenous than other countries reporting a larger deprivation gradient, and in such societies, structural racial differences may contribute more to poor COVID outcomes than elements of deprivation.

## Introduction

Initially called "The Great Leveller", 1918's influenza pandemic appeared to affect rich and poor equally. Subsequent research, however, showed a disproportionate impact on poorer populations [1]. More recently, seasonal influenza has mirrored this effect, but without yet revealing a causal relationship [2].

When the World Health Organisation declared a COVID-19 pandemic, a common refrain was that "we are all in this together" [3]. However, early reports showed greater mortality rates from COVID-19 in areas with lower income and higher unemployment [4]. In the United Kingdom, more deprived areas reported twice the mortality rate of affluent neighbourhoods [5] and similar patterns were observed in South America [6], and in the Asia-Pacific region [7].

Debate in the UK and US has focussed on ethnicity, rather than deprivation, as being the more critical determinant of COVID-19 in inequitable societies [8]. However, Ireland's population does not display the same level of ethnic diversity as other, larger countries. Other factors linked with COVID-19 incidence and mortality have been proposed, including age [9], comorbidity [10], urban density [11] and health-seeking behaviour [12].

The objective of this paper is to explore COVID-19 incidence and outcomes across three pandemic waves in the context of affluence and deprivation in Ireland using the *Pobal* Haase-Pratschke (HP) Relative Index [13], given Ireland's unique demographic structure.

## Methods

This is a retrospective cohort study of routinely notified COVID-19 cases confirmed by polymerase chain reaction (PCR), recorded on a standardised notification form with a list of possible comorbidities and recorded in the Computerised Infectious Disease Reporting system (CIDR) [14] managed by the Health Protection Surveillance Centre (HPSC), Ireland [15]. All cases were notified between 1st March 2020 and 13th May 2021.

Pandemic timing was represented in waves, based on rising or falling trends in national incidence rates: wave 1 from 1st March 2020 to 1st August 2020; wave 2 from 2nd August 2020 to 21st November 2020; and wave 3 from 22nd November 2020 to the end of the study period of 13th May 2021 [16]. This period was chosen to reflect the period before COVID vaccination was well established within the Irish population.

The deprivation score was based upon the 2016 *Pobal* HP Relative Deprivation Index, applied to the Central Statistics Office (CSO) national 2016 Census. The HP Index measures deprivation at the CSO small area level (approximately 100 households) [17] and identifies three dimensions: demographic profile; social class composition; and labour market situation. Each dimension is determined using routinely-collected Census data and the HP Index is derived from factor analysis.

The relative HP Index score was used in this study and cases were categorised from most deprived (HP1) to most affluent (HP5) based on standard deviations from the mean as shown in Fig 1. The HP relative deprivation index, in which affluence or deprivation is determined on the basis of distance in standard deviations from the mean, is an internationally recognised measure of deprivation and is used extensively within the public sector in Ireland [13]. Due to relatively small numbers, categories HP1 and HP2, and HP4 and HP5 were combined in the analysis to produce three categories: deprived (HP1 & HP2); average (HP3); and affluent (HP4 & HP5).

The small area HP Index score for each case was derived from the residential address from the CIDR database, using *Health Atlas Ireland*, interfaced with An Post's *GeoDirectory* [18] (www.healthatlasireland.ie). The automatic address matching process was supplemented by manual matching to achieve an overall small area geocoding rate of approximately 98%. Nursing home residents were classified by location of the home and occasionally the residence of occupational groups was recorded as the workplace. Where addresses were non-unique, a random neighbourhood match was used. This entailed linking the address to the closest identifiable address in the same geographic small area, meaning that the neighbourhood characteristics including deprivation score and population density can be applied to the non-unique address.

The outcome variables included: admission to hospital; admission to intensive care unit (ICU) and mortality as recorded on CIDR. COVID-19 mortality included cases with laboratory-confirmed COVID-19 infection; causality is not inferred [19].

Cases with an outbreak identifier were categorised as "outbreak-associated", and "travel-related" if that was the "most likely transmission source". "Comorbidity", a binary Yes/No variable, was documented as "Yes" if one or more underlying condition was recorded, and "No" if recorded as such or if left blank.

Statistical methods included: Pearson's chi-square test and particularly the G-test to determine Odds Ratios for multiple variables; Likelihood odds ratios with 95% C.I for multivariable analysis.; and Cuzick non-parametric test for trends across pandemic waves. For the purposes of modelling, variables were included which had been highlighted by other authors and which were significant at the $p = 0.10$ level during initial univariate analysis. A backward stepwise elimination process was followed and the model finally selected which returned the highest $R^2$ value. The CSO Census 2016 projected to 2020 (M2F2) [20] available on *Health Atlas Ireland* was used as the background population denominator [21]. Calculation of $R^2$ and effect size partial eta-squared ($\eta^2_p$) were included in order to illustrate the overall contribution of each variable to the change in outcome status. COVID-19 incidence was calculated by age group and by deprivation category. A rate ratio was determined to compare the more affluent and more deprived categories against the average deprivation value.

Given the legal obligation and operational imperatives of the HSE to protect the health and wellbeing of the general public during the pandemic in Ireland, Research Ethics Committee (REC) approval was not sought for this study. In compliance with the General Data Protection Regulations (GDPR), and the Health Research Regulations (HRR), an anonymised dataset was used. Funding was not required, or sought, for this study.

## Results

The first Irish case of COVID-19 in Ireland was notified on March 2nd 2020, with infections in wave 1 rising to a peak in April, falling to low levels during the summer (wave 2), rising again in autumn and into winter (wave 3) [22]. Wave 3 continued into the early summer of 2021.

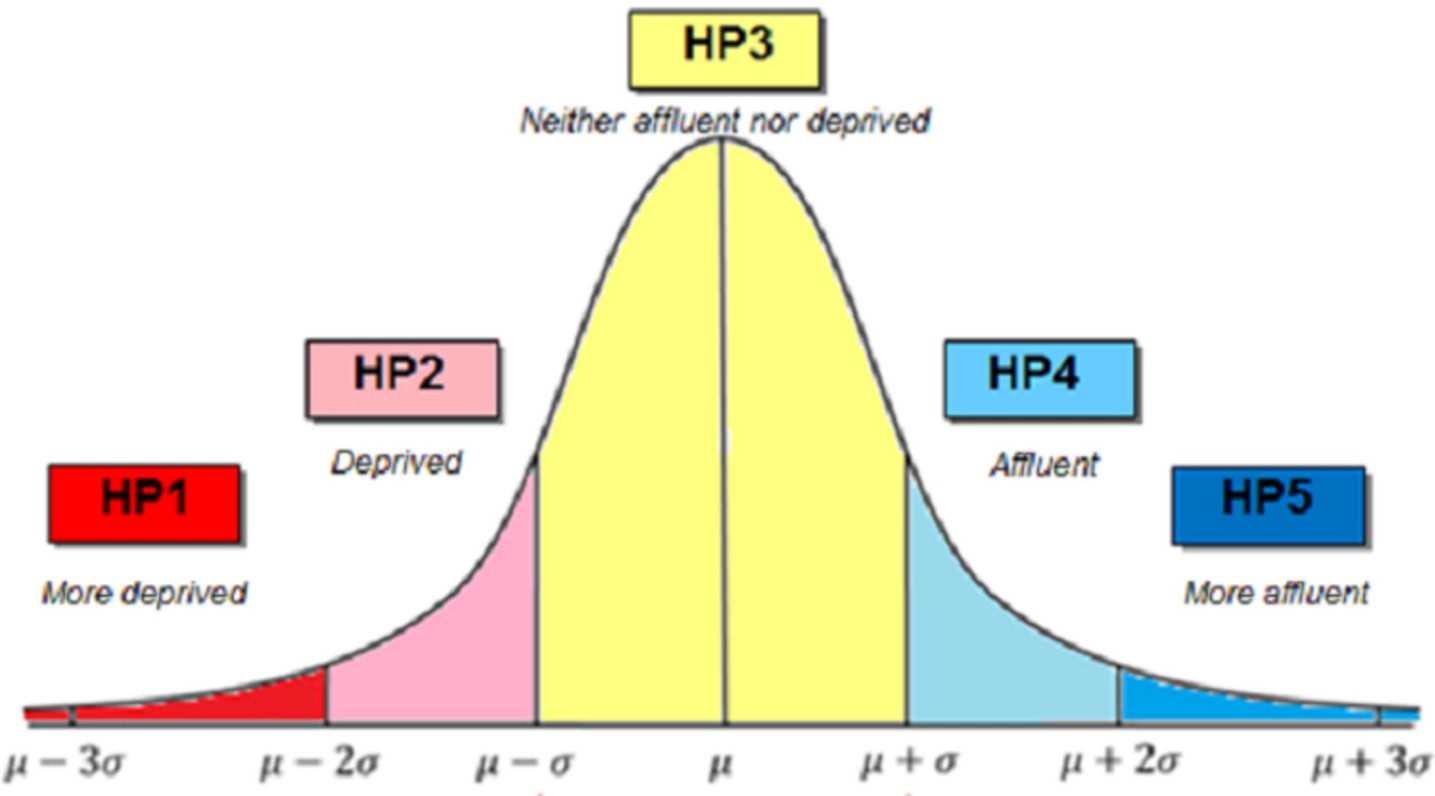

**Fig 1. HP index deprivation categories.**

By 13th May 2021, 252,637 PCR-confirmed cases were notified to HPSC. Of these, 14,420 (5.7%) were admitted to hospital; and of these admissions, 1,536 (0.6%) were admitted to ICU. There were 4,646 (1.8%) deaths among those diagnosed with COVID-19.

### Descriptive analysis

Table 1 summarises the distribution of key parameters of the study population by each HP category. Of the total cases, 131,515 (52.1%) were female, and 121,021 (47.9%) were male. The age distribution of cases was as follows: 136,784 (54.1%) were aged under 40 years of age; 38,867 (15.4%) aged 40 to 49 years; 34,112 (13.5%) aged 50–59 years; 19,037 (7.5%) aged 60–69 years; 11,283 (4.5%) aged 70–79 years and 12,513 (5.0%) were aged 80 years and over.

Approximately 80,155 cases (31.7%) were outbreak-associated and 3,631 (1.4%) were related to travel. Healthcare workers accounted for 11.1% of cases (28,041). Where information was available, 41,806 cases (18.1%) had at least one underlying condition.

Over the study period 25,147 cases (10.0%) occurred during wave 1, 43,560 (17.2%) during wave 2 and 183,930 (72.8%) during wave 3. As shown in Fig 2, the epidemic curve showed peaks in April 2020, October 2020 and January 2021. Throughout wave 1, only the wild virus type was present, and by wave 3, the alpha variant was dominant [23].

### Deprivation/affluence by incidence and risk ratio by COVID wave

As shown in Fig 3, the incidence of COVID-19 throughout the three pandemic waves was higher in the more deprived population, except among those aged 20 to 39 years, and this latter finding was consistent in each wave. In wave 1, possibly due to the "masking" effect of the 20

**Table 1. COVID-19 case parameters by HP Index category.**

| | HP1 Most deprived | | HP2 More deprived | | HP3 Average | | HP4 More affluent | | HP5 Most affluent | | Total |
|---|---|---|---|---|---|---|---|---|---|---|---|
| | **n** | **%** | **n** | **%** | **n** | **%** | **n** | **%** | **n** | **%** | **n** |
| Cases | **9,685** | **3.8** | **34,790** | **13.8** | **166,382** | **65.9** | **37,614** | **14.9** | **4,166** | **1.6** | **252,637** |
| *Age group* | | | | | | | | | | | *Missing n = 30* |
| 0–39 | 5,326 | 55.0 | 17,908 | 51.5 | 88,055 | 52.9 | 22,575 | 60.0 | 2,920 | 70.1 | 136,784 |
| 40–49 | 1,363 | 14.1 | 4,963 | 14.3 | 26,099 | 15.7 | 5,838 | 15.5 | 604 | 14.5 | 38,867 |
| 50–59 | 1,119 | 11.6 | 4,445 | 12.8 | 23,475 | 14.1 | 4,724 | 12.6 | 349 | 8.4 | 34,112 |
| 60–69 | 798 | 8.2 | 2,949 | 8.5 | 12,803 | 7.7 | 2,334 | 6.2 | 153 | 3.7 | 19,037 |
| 70–79 | 646 | 6.7 | 2,214 | 6.4 | 7,392 | 4.4 | 958 | 2.5 | 73 | 1.8 | 11,283 |
| 80+ | 646 | 6.7 | 2,214 | 6.4 | 7,392 | 4.4 | 958 | 2.5 | 73 | 1.8 | 12,513 |
| *Sex* | | | | | | | | | | | *Missing n = 105* |
| M | 4,423 | 45.7 | 16,401 | 47.2 | 79,764 | 48.0 | 18,410 | 49.0 | 2,023 | 48.6 | 121,021 |
| F | 5,260 | 54.3 | 18,378 | 52.8 | 86,559 | 52.0 | 19,179 | 51.0 | 2,139 | 51.4 | 131,515 |
| *Comorbidity (if any)* | | | | | | | | | | | *Missing n = 23,082* |
| Yes | 1,934 | 21.8 | 6,860 | 21.6 | 27,342 | 18.0 | 5,136 | 15.0 | 534 | 14.3 | 41,806 |
| No | 6,918 | 78.2 | 24,848 | 78.4 | 124,653 | 82.0 | 29,030 | 85.0 | 3,204 | 85.7 | 188,653 |
| *Healthcare worker* | | | | | | | | | | | *Missing n = 0* |
| Yes | 728 | 7.5 | 3,263 | 9.4 | 19,121 | 11.5 | 4,373 | 11.6 | 556 | 13.3 | 28,041 |
| Other* | 8,957 | 92.5 | 31,527 | 90.6 | 147,261 | 88.5 | 33,241 | 88.4 | 3,610 | 86.7 | 224,596 |
| *Outbreak associated* | | | | | | | | | | | *Missing n = 0* |
| Yes | 3,508 | 36.2 | 12,545 | 36.1 | 53,698 | 32.3 | 9,538 | 25.4 | 866 | 20.8 | 80,155 |
| No | 6,177 | 63.8 | 22,245 | 63.9 | 112,684 | 67.7 | 28,076 | 74.6 | 3,300 | 79.2 | 172,482 |
| *Travel related* | | | | | | | | | | | *Missing n = 0* |
| Yes | 80 | 0.8 | 298 | 0.9 | 2,269 | 1.4 | 811 | 2.2 | 173 | 4.2 | 3,631 |
| Other* | 9,605 | 99.2 | 34,492 | 99.1 | 164,113 | 98.6 | 36,803 | 97.8 | 3,993 | 95.8 | 249,006 |
| *Wave of pandemic* | | | | | | | | | | | *Missing n = 0* |
| 1 | 659 | 6.8 | 3,134 | 9.0 | 16,812 | 10.1 | 4,074 | 10.8 | 468 | 11.2 | 25,147 |
| 2 | 1,798 | 18.6 | 5,904 | 17.0 | 28,796 | 17.3 | 6,363 | 16.9 | 699 | 16.8 | 43,560 |
| 3 | 7,228 | 74.6 | 25,752 | 74.0 | 120,774 | 72.6 | 27,177 | 72.3 | 2,999 | 72.0 | 183,930 |
| *Hospital admission* | | | | | | | | | | | *Missing n = 0* |
| Yes | 737 | 7.6 | 2,618 | 7.5 | 9,326 | 5.6 | 1,633 | 4.3 | 106 | 2.5 | 14,420 |
| Other* | 8,948 | 92.4 | 32,172 | 92.5 | 157,056 | 94.4 | 35,981 | 95.7 | 4,060 | 97.5 | 238,217 |
| *ICU admission* | | | | | | | | | | | *,Missing n = 0* |
| Yes | 72 | 0.7 | 264 | 0.8 | 1,005 | 0.6 | 183 | 0.5 | 12 | 0.3 | 1,536 |
| Other* | 9,613 | 99.3 | 34,526 | 99.2 | 165,377 | 99.4 | 37,431 | 99.5 | 4,154 | 99.7 | 251,101 |
| *Mortality* | | | | | | | | | | | *Missing n = 0* |
| Yes | 225 | 2.3 | 828 | 2.4 | 3,204 | 1.9 | 369 | 1.0 | 20 | 0.5 | 4,646 |
| No | 9,460 | 97.7 | 33,962 | 97.6 | 163,178 | 98.1 | 37,245 | 99.0 | 4,146 | 99.5 | 247,991 |

* Includes "No" or "Not stated"

to 39 year cohort, there was no overall gradient of incidence with deprivation for all age groups combined.

The affluent/deprived rate ratio was found to follow the above pattern with a higher risk of the deprived being a COVID-19 case across all waves of the pandemic, and in all age groups with the exception of the 20 to 29 year age group, which showed a reversed pattern as seen in Fig 3.

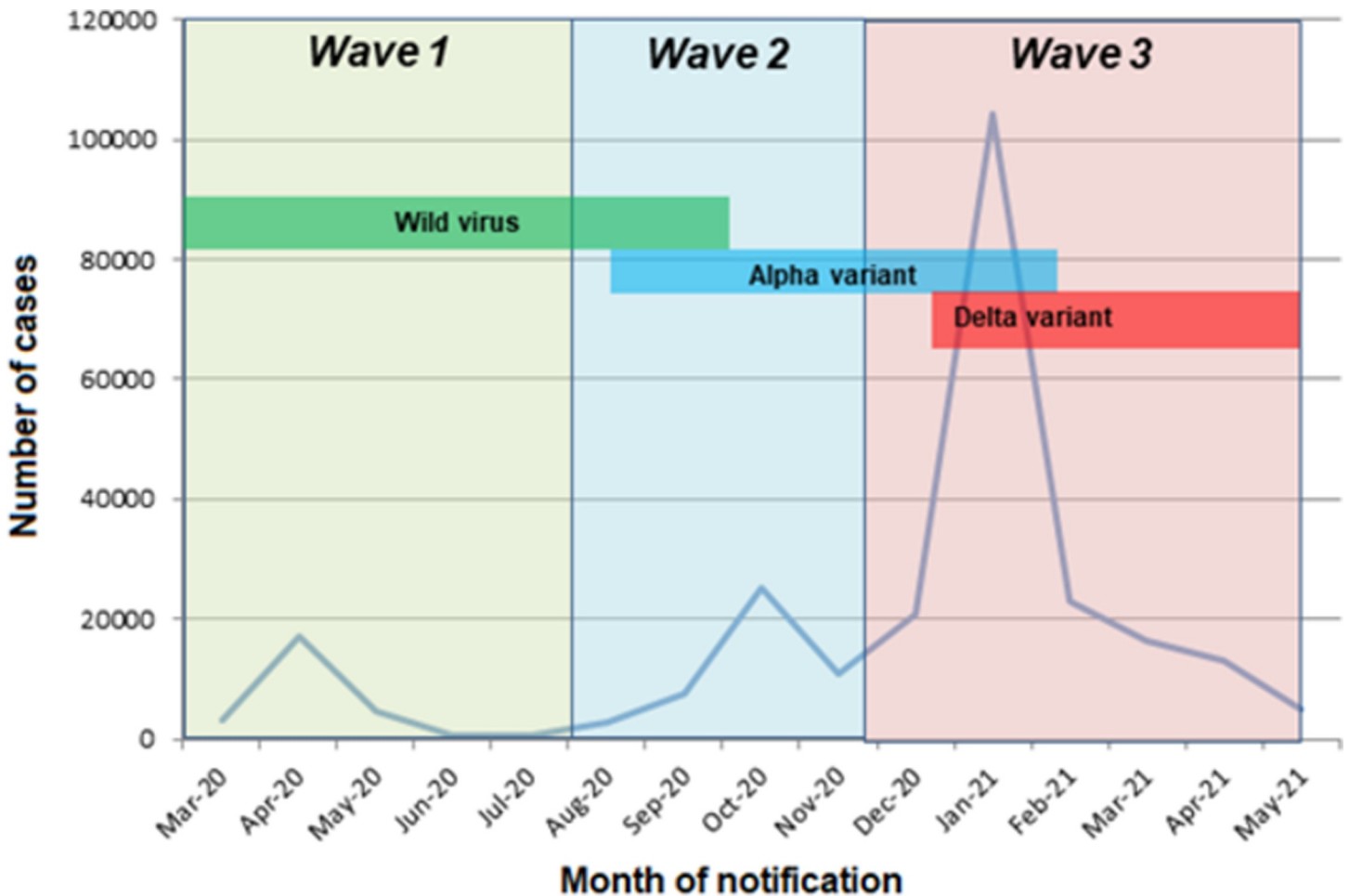

**Fig 2. Pandemic curve showing number of confirmed COVID-19 cases by pandemic wave and COVID-19 variant prevalence.**

The above findings were found to be most marked in the older age groups, i.e. above 60 years of age.

## Multivariable analysis

The following parameters were included in the multivariable analysis: HP category, age; gender; comorbidity; pandemic wave; outbreak-associated; and travel-related infection. Factors deemed to be significant at the 10% level ($p < 0.10$) in univariate analysis were included in building the final model. The method adopted was a backward elimination stepwise regression approach. Significant results are shown in Table 2 for the three outcomes: hospital admission, ICU admission and mortality. Variable interactions are included for age and deprivation.

The strongest predictor of ICU admission was comorbidity (aOR = 27.82, 95% CI 22.47–34.46), followed by increasing age up to the age of 80+ years (70–79 years aOR = 9.67; 95% CI 6.74–12.58). The next strongest predictor of ICU admission was whether the infection was associated with an outbreak (aOR = 1.30; 95% CI 1.14–1.49). Patients diagnosed in wave 2 of the pandemic were significantly less likely to be admitted to ICU (aOR = 0.63; 95% CI 0.50–0.78). Males were significantly less likely to be admitted to ICU (aOR = 0.54; 95% CI 0.48–0.62), as were cases with a history of travel (aOR = 0.69; 95% CI 0.47–0.99), although in the

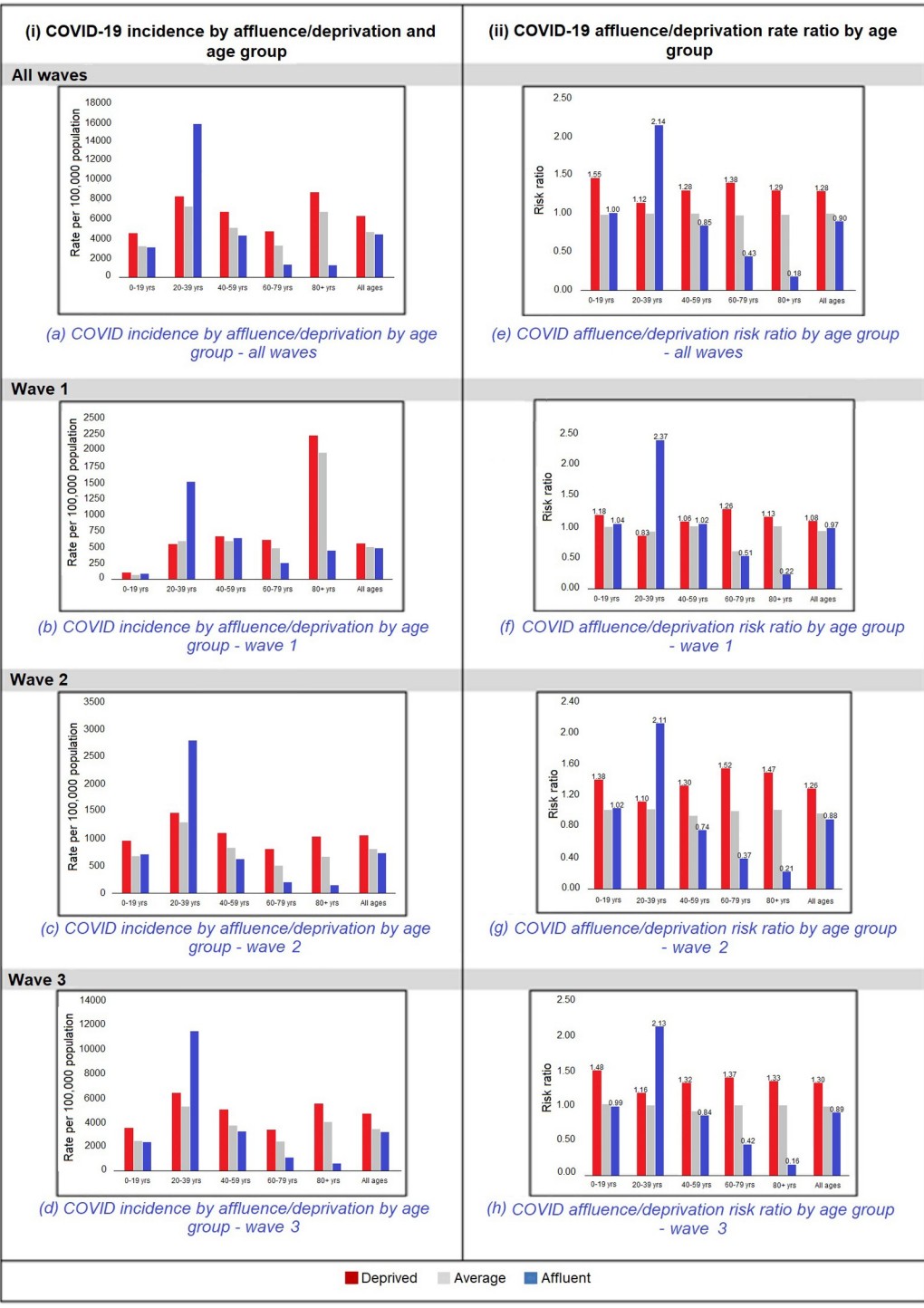

**Fig 3. COVID-19 incidence by affluence/deprivation and age group by pandemic wave and COVID-19 affluence/deprivation rate ratio by age group by pandemic wave.**

latter case, the effect size was not large. Deprivation was not associated with any difference in likelihood of ICU admission. Overall, this model accounted for 26.1% of the variation ($R^2$ = 0.2609) in the outcome (ICU admission).

**Table 2. Multivariable analysis by outcome–hospital admission, ICU admission and mortality.**

| Variable | Value | n | aOR | 95% CI | η² Main | η² Total | p-value |
|---|---|---|---|---|---|---|---|
| **Hospital admission (n = 14,420)** | | | | | | | |
| Age | 0–39 | 2,454 | 1.00 | | 0.590 | 0.730 | - |
| | 40–49 | 1,354 | 1.94 | (1.78, 2.10) | | | < 0.0001 |
| | 50–59 | 1,902 | 3.07 | (2.84, 3.31) | | | < 0.0001 |
| | 60–69 | 2,170 | 7.08 | (6.58, 7.62) | | | < 0.0001 |
| | 70–79 | 3,029 | 19.44 | (18.05, 20.92) | | | < 0.0001 |
| | 80+ | 3,693 | 23.93 | (22.30, 25.67) | | | < 0.0001 |
| Comorbidity | Yes | 7,689 | 3.27 | (3.11, 3.42) | 0.236 | 0.328 | 0.3319 |
| Wave of pandemic | 1 | 3,328 | 1.00 | (Ref) | 0.015 | 0.029 | - |
| | 2 | 1,821 | 0.67 | (0.62, 0.72) | | | 0.0001 |
| | 3 | 9,457 | 0.82 | (0.77, 0.88) | | | < 0.0001 |
| Outbreak-associated | Yes | 6,030 | 1.07 | (1.02, 0.12) | 0.002 | 0.004 | < 0.0078 |
| Deprivation | HP1 & HP2 | 3,355 | 1.22 | (1.15, 1.28) | | | < 0.0001 |
| | HP3 (average) | 9,326 | 1.00 | (Ref) | 0.008 | 0.027 | - |
| | HP4 & HP5 | 1,739 | 0.97 | (0.91, 1.04) | | | 0.3918 |
| Sex | Male | 7,702 | 1.32 | (1.26, 1.38) | 0.012 | 0.021 | 0.5455 |
| Interaction Deprivation*age | HP1 & HP2 | 3,355 | 1.26 | (1.21, 1.32) | | | < .0001 |
| | HP3 (average) | 9,326 | Ref | Ref | | | - |
| | HP4 & HP5 | 1,739 | 0.95 | (0.90, 1.01) | | | 0.099 |
| Model | | | | | | | $X^2(13) = 16680.95\ p< 0.0001$; Model $R^2$ value = 0.2109 |
| **ICU admission (n = 1,536)** | | | | | | | |
| Comorbidity | Yes | 1,396 | 27.82 | (22.47, 34.46) | 0.416 | 0.697 | < 0.0001 |
| Age | 0–39 | 133 | 1.00 | | 0.230 | 0.470 | - |
| | 40–49 | 205 | 5.38 | (4.1, 7.1) | | | < 0.0001 |
| | 50–59 | 315 | 9.69 | (7.5, 12.5) | | | < 0.0001 |
| | 60–69 | 453 | 26.70 | (26.7, 33.8) | | | < 0.0001 |
| | 70–79 | 367 | 36.37 | (28.3, 46.7) | | | < 0.0001 |
| | 80+ | 85 | 7.97 | (5.6, 11.3) | | | 0.0034 |
| Wave of pandemic | 1 | 438 | 1.00 | (Ref) | 0.013 | 0.036 | - |
| | 2 | 183 | 0.63 | (0.50, 0.78) | | | < 0.0001 |
| | 3 | 937 | 0.90 | (0.77, 1.05) | | | 0.2035 |
| Outbreak associated | Yes | 517 | 1.30 | (1.14, 1.49) | 0.01 | 0.029 | < 0.0001 |
| Travel associated | Yes | 32 | 0.69 | (0.47, 0.99) | 0.014 | 0.044 | 0.0481 |
| Deprivation | HP1 & HP2 | 336 | 1.07 | (0.92, 1.44) | | | 0.3703 |
| | HP3 (average) | 1,005 | 1.00 | (Ref) | 0.009 | 0.025 | - |
| | HP4 & HP5 | 195 | 0.93 | (0.76, 1.13) | | | 0.4567 |
| Sex | Male | 1,008 | 0.54 | (0.48, 0.62) | 0.035 | 0.1 | 0.8931 |
| Interaction Deprivation*age | HP1 & HP2 | 336 | 1.10 | (0.94, 1.28) | | | 0.22 |
| | HP3 (average) | 1,005 | 1 | | | | - |
| | HP4 & HP5 | 195 | 0.97 | (0.81, 1.15) | | | 0.70 |
| Model | | | | | | | $X^2(13) = 3457.89$; $p< 0.0001$; Model $R^2$ value = 0.2609 |
| **Mortality (n = 4,646)** | | | | | | | |
| Age | 0–39 | 30 | 1.00 | | 0.370 | 0.750 | - |
| | 40–49 | 57 | 6.80 | (3.94, 11.77) | | | < 0.0001 |
| | 50–59 | 152 | 21.19 | (13.00, 34.54) | | | < 0.0001 |
| | 60–69 | 437 | 109.65 | (69.07, 174.08) | | | < 0.0001 |
| | 70–79 | 1,121 | 403.70 | (255.68, 637.41) | | | < 0.0001 |
| | 80+ | 2,886 | 932.53 | (593.96, 1464.07) | | | < 0.0001 |
| Comorbidity | Yes | 4,123 | 9.44 | (8.29, 10.76) | 0.149 | 0.445 | < 0.0001 |
| Wave of pandemic | 1 | 1,530 | 1.00 | (Ref) | 0.026 | 0.069 | - |

(Continued)

**Table 2.** (Continued)

| Variable | Value | n | aOR | 95% CI | η² Main | η² Total | p-value |
|---|---|---|---|---|---|---|---|
| | *2* | 386 | 0.44 | (0.38, 0.51) | | | < 0.0001 |
| | *3* | 2,767 | 0.89 | (0.81, 0.98) | | | 0.0237 |
| Outbreak associated | *Yes* | 3,083 | 1.73 | (1.58, 1.89) | 0.015 | 0.048 | < 0.0001 |
| Travel associated | *Yes* | 11 | 0.45 | (0.23, 0.85) | 0.030 | 0.109 | 0.0146 |
| Sex | *Male* | 2,486 | 0.83 | (0.74, 0.92) | 0.014 | 0.044 | 0.0006 |
| Deprivation | *HP1 & HP2* | 1,053 | 0.93 | (0.85, 1.02) | | | 0.1382 |
| | *HP3 (average)* | 3,204 | 1.07 | (Ref) | 0.005 | 0.015 | - |
| | *HP4 & HP5* | 389 | 0.83 | (0.71, 0.97) | | | 0.0146 |
| Interaction Deprivation*age | *HP1 & HP2* | 1,053 | 1.23 | (1.01, 1.51) | | | 0.05 |
| | *HP3 (average)* | 3,204 | 1 | | | | |
| | *HP4 & HP5* | 389 | 0.81 | (0.64, 1.04) | | | 0.10 |
| Model | | | | $X^2(13) = 15933.05; p< 0.0001;$ Model $R^2$ value = 0.4863 | | | |

Admission to hospital was strongly associated with: increasing age (age 70–79 years aOR = 13.07; 95% CI 12.14–14.06; age 80+ years aOR = 13.26; 95% CI 12.31–14.27); and comorbidity (aOR = 3.27; 95% CI 3.11–3.42). Cases diagnosed in wave 2 or 3 of the pandemic were less likely to be admitted to hospital (wave 2 aOR = 0.67; 95% CI 0.62–0.72; wave 3 aOR = 0.82; 95% CI 0.77–0.88). Admission risk was also positively associated with increasing deprivation (HP1 & HP2 aOR = 1.22; 95% CI 1.15–1.28), but there was no association with increasing affluence (HP4 & HP5 aOR = 0.97; 95% 0.91–1.04). Male gender was associated with increased risk of hospital admission (aOR = 1.32; 95% CI 1.26–1.38). Overall, this model accounted for 21.1% of the variation ($R^2$ = 0.2109) in the outcome (hospital admission).

Case mortality showed a steadily increasing risk up to the age of 80+ (aOR = 494.6, 95% C.I. 305.56–800.59). Case mortality was strongly associated with the presence of one or more comorbidities (aOR = 9.44, 95% C.I. 8.29–10.76). Mortality was highest in wave 1 when compared to wave 2 (aOR = 0.44; 95% CI 0.38–0.51) or wave 3 (aOR = 0.89; 95% CI 0.81–0.98). Outbreak-associated cases also had a higher mortality (aOR = 1.73; 95% CI 1.58–1.89). Patients contracting COVID-19 as a result of travel were less likely to die (aOR = 0.45; 95% CI 0.23–0.85), as were males (aOR = 0.83; 95% CI 0.74–0.92), although in the latter case, the effect size was smaller. The risk of death was slightly less in the more affluent category (HP4 & HP5 aOR = 0.83; 95% CI 0.71–0.97) but there was no association in the more deprived category (HP1 & HP2 aOR = 0.93; 95% CI 0.85–1.02).

Because increasing age can be associated with increasing deprivation, the age term shown in Table 2 is corrected for interaction with deprivation. For purposes of space, the uncorrected term is not shown. While some interaction was observed between age and deprivation, this was only significant for the outcome of hospital admission.

For purposes of space, age is shown only as an interaction term with deprivation. This was done to try and consider whether the impact of age on all three outcomes could be related more to increasing deprivation in the older age groups. However, when compared directly, there was a certain interaction between age and deprivation, but this was only significant for hospital admission. Otherwise, age remains an independently high predictor of all three outcomes, and indeed is the greatest predictor for hospital admission and for mortality, and is second only to comorbidity for ICU admission.

## Discussion

This study is the first in Ireland to explore COVID-19 by small geographic area as a function of affluence/deprivation using the HP index across three pandemic waves. Of the total number of cases, 98.4% were geo-referenced to a small area. Our study showed that the risk of being a confirmed case of COVID-19 is higher among the more deprived categories for all ages across

all three pandemic waves, except for the 20–39 years age category, in whom the reverse pattern was found.

This latter finding may be associated with a high proportion (46%) of cases in the 20–39 age category being health care workers, who had both an increased occupational exposure to COVID-19 and a requirement for serial testing, leading to possible ascertainment bias. Ascertainment bias due to increased testing for travel purposes may also play a part, as the 20–39 age group is more associated with travel. This finding, however, has not been reported internationally. Typically, those professions either with greater public contact or for whom working from home was not an option were considered to be more at risk of COVID-19, in the pre-vaccine era. Those from more affluent professions, for whom social distancing and working from home were a realistic option, had a greater degree of protection during this time [24].

The main determinants of hospital admission were: increasing age, comorbidity, a positive COVID-19 diagnosis in an earlier wave of the pandemic, male sex and being resident in a more deprived small area. By contrast, the main determinants for ICU admission and death were: comorbidity, increasing age, association with an outbreak and COVID-19 diagnosis during an earlier wave of the pandemic. No convincing deprivation gradient was found for ICU admission, but mortality was slightly lower in the more affluent population. While there was some degree of variable interaction between age and deprivation, age remained, even after correction, as one of the strongest predictors of hospital admission, ICU admission and death from COVID.

Those patients diagnosed in later waves were likely to have benefitted from the experience and confidence that grew among healthcare providers. Additionally, diagnostic bias in the earliest wave may have resulted in less severe cases not being recognised or diagnosed. Along with the provision of supports for quarantining and isolation, it is likely that cases were able to be managed at lower levels of complexity and without hospital admission [25]. Additionally, with increasingly focussed testing criteria, a greater number of less severe cases are likely to have been identified later in the pandemic.

While deprivation is closely associated with poorer outcomes for many health conditions, that pattern was not observed consistently in our study. This finding is echoed in a recent study of trends in mortality rates over a five-month period in England, which found that deprivation and ethnic group were not associated with death among hospitalised patients once appropriately adjusted for risk. The authors note that their finding was consistent with other US studies which suggest that the widely reported effect of deprivation and ethnicity is most likely due to differences in exposure to the disease [26]. Ireland is ethnically more homogenous than the UK and the US and removing ethnicity as a factor in terms of COVID outcomes may have reduced the overall impact of deprivation in an Irish population.

Deprivation is a complex concept, and the Irish HP Index is a sensitive, multi-dimensional tool. Our findings suggest that once adjustment is carried out for factors that are confounders for deprivation–age and comorbidity in this case–the overall impact of deprivation is reduced. The lack of a clear deprivation gradient for COVID-19 outcomes suggests equitable access and provision of COVID-19 related health care across all strata of Irish society. Internationally, however low income [27] and overcrowding [28] have been associated with poor outcomes from COVID-19.

The apparent divergence of Irish findings may be explained by the presence of significant ethnic minorities in other countries, perhaps confounded by deeper inequities in these societies. Ethnicity is not included in the Irish HP Index, nor is it mentioned in other international measures of deprivation [29–31].

It must be noted, however, that recent Irish research has noted a link between COVID-19 incidence and deprivation [32]. In Europe, association between deprivation and COVID

admission and mortality has been determined in Northern Ireland [33], England [34, 35], Scotland [36], France [37] and Spain [38], among others. However, it is not necessarily simple, or linear. Germany, for example, reported that earlier cases tended to be more affluent, followed by an increasing likelihood of illness in the more deprived [39]. This was also determined in early Irish observations [40] and suggests additional complexity that has not previously been considered. In addition, Italian researchers determined that the deprivation profile of cases changed after lockdown, and pointed out the differential impact of lockdown on people in different deprivation categories [41].

This study was one of secondary data analysis, that is, it analysed data that had been collected for another purpose, namely COVID-19 surveillance for Public Health protection and infection control. We used literature sources to identify those factors which were deemed to have the greatest impact on COVID outcome. As a result, our choice of variables was limited to those identified as being important for the purposes of surveillance. It is possible, therefore, that we may not have therefore been able to pick up on nuances that would otherwise be available to researchers collecting data in a less emergent setting.

Critically, this study was performed using data from the pre-vaccine phase of the COVID pandemic in Ireland. We believed that in order to determine whether any deprivation gradient existed in an Irish population with respect to coronavirus infection, we had to look at pre-immunisation data. The reason for this is that the COVID vaccine was made available to all individuals, free of charge, and on the basis of risk. It therefore sought to reduce any socio-demographic inequalities that may have existed. By including waves subsequent to the vaccine roll-out, we believe that any nuance or gradient that may have existed would have been nullified. We therefore wanted to know whether, and the extent to which, social deprivation would prove to be a risk factor in an Irish population for future pandemic planning.

Data quality overall, however, was a strength in this study, because it was derived from a single source, namely the Computerised Infectious Disease Reporting (CIDR) system governed by the Health Protection Surveillance Centre [15]. CIDR includes all hospital- and community-based, laboratory-confirmed COVID-19 cases in Ireland, and is the "source of truth" for all national COVID-19 reporting processes. Clinically suggestive but PCR-negative cases of COVID-19 were not included in our study, providing for a more specific but perhaps less inclusive case definition.

A number of potential causes of bias were considered. COVID-19 testing is free to all in Ireland and it is unlikely that test uptake was influenced by deprivation. Despite address quality issues, over 98% of cases were successfully geo-referenced, and was not considered a source of bias in deprivation categorisation.

Other studies have examined age as a continuous variable, whereas in our study, we used age categories. We are aware in doing so that we may have lost some power, but these age categories also strongly reflect decision making in a clinical setting in Ireland, and we wished our analysis to maintain the same clinical significance. In addition, when we included age as a continuous variable in the model, the resulting lower Odds Ratios made it difficult to represent fully the changing impact on outcome at different ages. In other words, we believe that there was a non-linear relationship between age and outcomes. Finally, when we included age as a continuous variable, the resulting $R^2$ was lower than when it was modelled as a categorical variable.

The degree of asymptomatic carriage of COVID-19, or false-negative testing may have resulted in bias. These are difficult to determine and validations were not carried out for this study. Therefore the definition of a case as a person with laboratory-confirmed infection will not include persons who were ill and who returned a negative test, or those who were asymptomatic and did not seek a test.

Missing data for the "Comorbidity" variable must be balanced against the very high odds ratios we determined for all three of the outcomes in this study and the fact that international literature consistently cites comorbidity as a predictive factor in COVID-19 outcomes [10].

## Conclusion

Our study found a deprivation gradient in COVID-19 incidence (with the exception of the 20–39 age group) and in hospital admission, but no similar trend in relation to ICU admission or mortality. Deprivation, however, is a multi-factorial concept, and removing some of these factors by statistical adjustment helps to illustrate which variables have the greatest impact. This study echoed international literature in determining that the principal determinants of all outcomes (hospital admission, ICU admission and mortality) were age, comorbidity and diagnosis at an early wave of the pandemic.

Our results suggest a "non-democratic" transmission of COVID-19 in the Irish population, but this did not translate into poorer outcomes. Demographic differences, particularly relating to ethnicity, may help to explain the disparity between Irish and international experience.

Our study suggests that COVID-19 spreads easily through all strata of society and particularly in the more deprived population. However, in terms of more serious outcomes from COVID-19 in Ireland, our study suggests that there would appear to be truth in the saying that "we are all in this together".

## Author Contributions

**Conceptualization:** Angela McCourt, Anne O'Farrell, Fionnuala Donohue, Laurin Grabowsky, Paul Kavanagh, Patricia Garvey, Joan O'Donnell, Lois O'Connor, Matt Robinson, Howard Johnson.

**Data curation:** Declan McKeown, Angela McCourt, Anne O'Farrell, Laurin Grabowsky, Patricia Garvey, Joan O'Donnell, Lois O'Connor.

**Formal analysis:** Declan McKeown, Angela McCourt, Louise Hendrick, Anne O'Farrell, Laurin Grabowsky, Patricia Garvey, Joan O'Donnell, Howard Johnson.

**Investigation:** Angela McCourt, Louise Hendrick, Anne O'Farrell, Fionnuala Donohue, Laurin Grabowsky, Patricia Garvey, Joan O'Donnell, Howard Johnson.

**Methodology:** Louise Hendrick, Anne O'Farrell, Fionnuala Donohue, Laurin Grabowsky, Patricia Garvey, Joan O'Donnell, Anthony Staines, Howard Johnson.

**Project administration:** Declan McKeown, Fionnuala Donohue, Joan O'Donnell, John Cuddihy, Declan O'Reilly.

**Resources:** Patricia Garvey, Joan O'Donnell, Lois O'Connor, John Cuddihy, Matt Robinson, Declan O'Reilly.

**Software:** Anne O'Farrell, Laurin Grabowsky.

**Supervision:** Declan McKeown, Fionnuala Donohue, Paul Kavanagh, John Cuddihy, Declan O'Reilly, Anthony Staines, Howard Johnson.

**Validation:** Laurin Grabowsky, Paul Kavanagh, Lois O'Connor, John Cuddihy, Matt Robinson, Anthony Staines.

**Visualization:** Lois O'Connor, Matt Robinson, Anthony Staines, Howard Johnson.

**Writing – original draft:** Declan McKeown, Angela McCourt, Anne O'Farrell, Fionnuala Donohue, Laurin Grabowsky, Patricia Garvey, Joan O'Donnell, Howard Johnson.

**Writing – review & editing:** Declan McKeown, Louise Hendrick, Anne O'Farrell, Paul Kavanagh, Joan O'Donnell, Lois O'Connor, Anthony Staines, Howard Johnson.

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
