## [Decision Letter · Decision Letter 0]

17 Oct 2022

PONE-D-22-14948COVID-19 incidence and outcome by affluence/deprivation across three pandemic waves in Ireland: a retrospective cohort study using routinely collected dataPLOS ONE

Dear Dr. McKeown,

Thank you for submitting your manuscript to PLOS ONE. After careful consideration, we feel that it has merit but does not fully meet PLOS ONE’s publication criteria as it currently stands. Therefore, we invite you to submit a revised version of the manuscript that addresses the points raised during the review process. Please address these issues and others raised in the reviews.The modeling could be made more clear and additional models considered.Please address the selection process for the predictors included in the model? Were there other predictors available in the data source but not used in the model? Why did you categorize age in the modeling? Why not just use splines and model the effect of age as a continuous predictor? There is the potential to really lose information by categorizing. Please justify why those categories were chosen. Similarly, why was the HP index categorized? Are these standard categorizations or selected based on the data in this study? Could a continuous version, perhaps with splines, be used in the regression?Please submit your revised manuscript by Dec 01 2022 11:59PM. If you will need more time than this to complete your revisions, please reply to this message or contact the journal office at plosone@plos.org. Please include the following items when submitting your revised manuscript:A rebuttal letter that responds to each point raised by the academic editor and reviewer(s). You should upload this letter as a separate file labeled 'Response to Reviewers'.A marked-up copy of your manuscript that highlights changes made to the original version. You should upload this as a separate file labeled 'Revised Manuscript with Track Changes'.An unmarked version of your revised paper without tracked changes. You should upload this as a separate file labeled 'Manuscript'.

We look forward to receiving your revised manuscript.

Kind regards,

Edward Jay Trapido, ScD

Academic Editor

PLOS ONE

3. Please upload a copy of Figure 4, to which you refer in your text on page 7. If the figure is no longer to be included as part of the submission please remove all reference to it within the text.

Reviewers' comments:

Reviewer's Responses to Questions

**Comments to the Author**

1. Is the manuscript technically sound, and do the data support the conclusions?

Reviewer #1: Yes

Reviewer #2: Partly

2. Has the statistical analysis been performed appropriately and rigorously? 

Reviewer #1: Yes

Reviewer #2: Yes

3. Have the authors made all data underlying the findings in their manuscript fully available?

Reviewer #1: Yes

Reviewer #2: Yes

4. Is the manuscript presented in an intelligible fashion and written in standard English?

Reviewer #1: Yes

Reviewer #2: Yes

5. Review Comments to the Author

Reviewer #1: The present study conducted two major analyses: (1) calculation of COVID-19 incidence per population by deprivation and age group, and (2) multivariate logistic regression of three outcome measures to detect the association with deprivation among COVID-19 cases.

The two-part composition is the uniqueness of this manuscript. Thus, revisions would be recommended to make it clear for readers.

Abstract

Page 2 lines 29-32: Additional explanation may be inserted into second paragraph, to mention about calculation of COVID-19 incidence.

Page 2 lines 37-43: Forth paragraph (discussion/conclusion) appears to be a bit long; last sentence might be omitted.

Introduction section is concise and condensed.

Methods

Page 3 lines 68-70: the rationales of pandemic timing may need elaboration. For example, is this based on the effective reproduction number (Rt) or other measures suggesting declining/increasing trends?

It would also be noted whether vaccination was launched in early 2021 (Wave 3) in Ireland. It may be relavant to some outcome measures such as admission to an ICU and death.

Page 4 lines 87-88: a brief explanation may be added to "a random neighbourhood match".

Page 4 lines 95-101: description of statistical methods would be restructured in order to indicate the aim (to examine XXX) of each procedure.

Results

Page 7 lines 139-148: it is not sure whether any test statistics are available to judge "higher" incidence and risk ratio.

Page 10 lines 170-178: criteria for the effect size would be declared in Methods section in addition to Table 2.

Discussion

Page 10 line 191-page 11 line 201: some speculation may be added regarding why high incidence was observed among the more deprived categories, such as occupation-associated risk or hygiene habits?

Page 11 lines 215-222: again, vaccination rollout would be discussed as it could reduce hospitalization/severe symptoms and mortality.

Conclusion

Page 13 lines 257-263: it is recommended that the conclusion section does not include citations. Second paragraph might be moved to Discussion section.

Reviewer #2: General Comments:

Overall an interesting manuscript that uses administrative data sources to examine the association of material deprivation with COVID-19 incidence and outcomes (hospital admission, ICU admission, mortality). Literature review is adequate. The sample size is large and population-based. The methodology appears appropriate. The conclusions drawn are reasonable.

Major Comments:

The modeling could be made more clear and additional models considered. First, I don’t see how there is any forward/stepwise regression done as all that is reported seems to include all of the terms. What there some sort of selection process for the predictors included in the model? Where there other predictors available in the data source but not used in the model? They authors need to explain a bit more about what predictors were available and which were ultimately chosen. I personally also like to see unadjusted (bivariable) results (i.e., regression with individual predictors) so that I can see how inclusion of all predictors has changed some of the estimates.

Age is a very important predictor of death and likely other outcomes. Why did the authors chose to categorize age in the modeling? Why not just use splines and model the effect of age as a continuous predictor? There is the potential to really lose information by categorizing. Also if categorizing, would have to justify why those categories were chosen. Similarly, why was the HP index categorized? Are these standard categorizations or selected based on the data in this study? Could a continuous version, perhaps with splines, be used in the regression?

I like Figure 3 very much and it seems to suggest important interactions between age and deprivation and perhaps age, deprivation, and wave. Were interactions considered in the model? It would seem that some should be given the exploratory data analysis.

Minor Comments:

1. How were the 3 waves defined?

2. HP Index: Does the HP index cover the entire population? Are there any subgroups that wouldn’t apply? There is a suggestion that perhaps people living in a continuing care setting may not be covered. More importantly, what are the properties of the deprivation index. The authors could do more to explain what the different dimensions mean and what the relationship to COVID-19 could be.

3. Is the unit of analysis the patient/individual? Is it possible to have more than one infections for a person added to the administrative data source?

4. How many hospitals are there in the study population? Is there variation in admission standards to hospital (and ICU)? If so, a random effect for hospital may need to be considered in the modeling to account for correlation in patients/outcomes in the same hospital.

5. The data collection process could be made more clear. Is there, say, a 1 page form for reporting? Are comorbidities just gleaned from that form or could comorbidities be determined from another data source? Is there data available about the type(s) of comorbidities? It is not clear how much missing and/or misclassification is present within the comorbidity variable.

6. The outcomes are not modeled jointly, so I would not refer to this as a multivariate regression but rather multivariable regression. A multivariate analysis is not really done in this manuscript.

7. Table 1. Seem to be missing a label under health worker: ie there are 2 yes and 2 no with that variable.

8. Figure 1 and think it would be good to actually show the distribution of HP index for the study population. Further, the same kind of graphic could be done for each of the 3 waves.

9. The justification for not requiring an Ethics Review is not clear to me.

6. PLOS authors have the option to publish the peer review history of their article (what does this mean?). If published, this will include your full peer review and any attached files.

Reviewer #1: No

Reviewer #2: No

---

## [Author Response · Author response to Decision Letter 0]

27 Mar 2023

Reviewer #1

The present study conducted two major analyses: (1) calculation of COVID-19 incidence per population by deprivation and age group, and (2) multivariate logistic regression of three outcome measures to detect the association with deprivation among COVID-19 cases. The two-part composition is the uniqueness of this manuscript. Thus, revisions would be recommended to make it clear for readers.

Abstract

Comment: Page 2 lines 29-32: Additional explanation may be inserted into second paragraph, to mention about calculation of COVID-19 incidence.

Response: Additional comment added about notification to the national Computerised Infectious Diseases Register (CIDR)

Comment: Page 2 lines 37-43: Fourth paragraph (discussion/conclusion) appears to be a bit long; last sentence might be omitted.

Response: Last sentence has been omitted.

Introduction 

Comment: Section is concise and condensed.

Methods

Comment: Page 3 lines 68-70: the rationales of pandemic timing may need elaboration. For example, is this based on the effective reproduction number (Rt) or other measures suggesting declining/increasing trends?

Response: Agree – the timing is based on rising and falling crude incidence at a national level and agreed by the Health Protection Surveillance Centre. Clarification has been added to text.

Comment: It would also be noted whether vaccination was launched in early 2021 (Wave 3) in Ireland. It may be relevant to some outcome measures such as admission to an ICU and death.

Response: Vaccination in Ireland only became established towards the end of the third wave, and while it may have contributed to the slowing off of infections in the summer of 2021, uptake only rapidly took off after the end of the third wave. Vaccination onset acknowledged in the text.

Comment: Page 4 lines 87-88: a brief explanation may be added to "a random neighbourhood match".

Response: The Eircode (postal code) system in Ireland was not fully embedded before the COVID pandemic hit. Databases had not been fully updated to reflect the new postal codes, which meant that patient addresses had to be recorded in long form. Because of the large rural population, many addresses were by townland only, meaning that a number of them are non-unique. For our purposes, we did not require x,y coordinates, and instead, non-unique addresses were linked to the closest geographic Small Area (an SA comprising 50-100 households). The SA is the smallest boundary available and each has a calculated population density and deprivation index. Therefore, the analysis did not require precise x,y location, but rough neighbourhood matching allowed us to identify the SA in which the address should be located.

Comment: Page 4 lines 95-101: description of statistical methods would be restructured in order to indicate the aim (to examine XXX) of each procedure.

Response: This has been done

Results

Comment: Page 7 lines 139-148: it is not sure whether any test statistics are available to judge "higher" incidence and risk ratio.

Response: The analysis is made on the basis of the error bars shown underneath the OR in figures 2 (e), (f), (g) and (h).

Comment: Page 10 lines 170-178: criteria for the effect size would be declared in Methods section in addition to Table 2.

Response: Criteria have been included in the methods section.

Discussion

Comment: Page 10 line 191-page 11 line 201: some speculation may be added regarding why high incidence was observed among the more deprived categories, such as occupation-associated risk or hygiene habits?

Response: A brief explainer has been added, with reference.

Comment: Page 11 lines 215-222: again, vaccination rollout would be discussed as it could reduce hospitalization/severe symptoms and mortality.

Response: Very good point. A paragraph has been added to reflect this. 

Conclusion

Comment: Page 13 lines 257-263: it is recommended that the conclusion section does not include citations. Second paragraph might be moved to Discussion section.

Response: Agree and done.

Reviewer #2:

General Comments:

Overall an interesting manuscript that uses administrative data sources to examine the association of material deprivation with COVID-19 incidence and outcomes (hospital admission, ICU admission, mortality). Literature review is adequate. The sample size is large and population-based. The methodology appears appropriate. The conclusions drawn are reasonable.

Major Comments:

Comment: The modeling could be made more clear and additional models considered. First, I don’t see how there is any forward/stepwise regression done as all that is reported seems to include all of the terms. What there some sort of selection process for the predictors included in the model? Where there other predictors available in the data source but not used in the model? They authors need to explain a bit more about what predictors were available and which were ultimately chosen. I personally also like to see unadjusted (bivariable) results (i.e., regression with individual predictors) so that I can see how inclusion of all predictors has changed some of the estimates.

Response: We adopted a backward elimination stepwise regression approach. If a variable was deemed to be significant at p<0.1 at the univariate level, it was included in the overall model. Therefore the variables included in the final model were those that were significant at this level in the univariate phase of analysis. This has been clarified in the most recent submitted manuscript.

The predictors available for the model were limited to the data captured. Given that the variables were derived from surveillance data, we were limited by what was deemed significant for data capture during a pandemic. As a result, certain nuances may have been lost which would otherwise be available to researchers collecting de novo data in a more elective setting. We decided to single out for specific attention variables which had been indicated as having clinical or statistical significance in other studies. This has been further clarified in the text. 

Comment: Age is a very important predictor of death and likely other outcomes. Why did the authors choose to categorize age in the modeling? Why not just use splines and model the effect of age as a continuous predictor? There is the potential to really lose information by categorizing. Also if categorizing, would have to justify why those categories were chosen. Similarly, why was the HP index categorized? Are these standard categorizations or selected based on the data in this study? Could a continuous version, perhaps with splines, be used in the regression?

Response: We are aware that in categorising age, we may have lost some power. However in doing so, and in selecting the chosen categories, we believed that we were reflecting other Irish analysis that seemed to determine differences in outcome in those age groups. When we used age as a continuous variable, our R-square was lower, and we believe that this may cloud the clinical impact that age has had on the Irish population. Therefore, we believed that in determining odds or risk differences by one year age unit could conceal the very real clinical impact that COVID has had on an Irish population. We felt that it was easier, when communicating risk, to do so using the age group approach, rather than run the risk of diluting the message, which we felt was important to share with our clinical and Public Health colleagues, both in Ireland and overseas. The COVID pandemic has already suffered as a result of public disinformation, and we wanted our message to be as clear as possible.

The Pobal HP Deprivation Index was generated for the Irish population by Trutz Haase and Jonathan Pratschke. For the purposes of determining material deprivation, the relative HP index is used as a five-category measurement that is based on standard deviations from a mean. The HP Index is recalculated at every Census cycle. The categorisations are therefore standard and will appear as the same categories throughout much of the public sector publications addressing deprivation in Ireland. We believe that a continuous version, because of the relatively small numbers involved in Ireland in comparison with other countries, would not have had the sensitivity to show deprivation differences by the time sub-analyses were being performed, and particularly for the multivariate calculations.

Comment: I like Figure 3 very much and it seems to suggest important interactions between age and deprivation and perhaps age, deprivation, and wave. Were interactions considered in the model? It would seem that some should be given the exploratory data analysis.

Response: Interactions were considered in the model, particularly interaction between deprivation and age, and that is controlled for in the model.

Minor Comments:

Comment 1. How were the 3 waves defined?

Response: The three waves were defined on the basis of increasing and decreasing crude COVID-19 incidence rates in the national population. The timing of the waves have been standardised by the national Health Protection Surveillance Centre in Dublin and are used commonly throughout all Irish government agencies when describing the pandemic timing

Comment 2. HP Index: Does the HP index cover the entire population? Are there any subgroups that wouldn’t apply? There is a suggestion that perhaps people living in a continuing care setting may not be covered. More importantly, what are the properties of the deprivation index. The authors could do more to explain what the different dimensions mean and what the relationship to COVID-19 could be.

Response: We felt that a more in-depth description of the HP index was beyond the scope of this paper. However, the source that is cited in the reference section (Haase T. The 2016 Pobal HP Deprivation Index (SA), Reference 13) states that the HP Index uses factor analysis to model three domains, each with a number of indicators, as shown below.

(1) Demographic profile

Rural deprivation is often the result of agricultural underemployment and/or emigration (which has been a longstanding cultural phenomenon in Ireland; the population has still not regained its 1841 population of 8 million, following the disastrous famines and emigrations of the later 1840s). Emigration from deprived rural areas is often the result of a mismatch between education, skills levels and expectations on the one hand, and on the other, with available job opportunities. Emigration is also socially selective, meaning that those who are of working age with skills leave the area, resulting in those left behind being even more economically dependent. Erosion of local workforce means that an area is less attractive for economic investment 

There are six indicators within the domain of demographic profile

• the percentage change in population over the previous five years (positive association);

• the percentage of population aged under 15 or over 64 years of age (negative association);

• the percentage of population with a primary school education only (negative association);

• the percentage of population with a third level education (positive association);

• the percentage of households with children aged under 15 years and headed by a single parent (positive association);

• the mean number of persons per room (positive association).

(2) Social class composition

Social Class Composition is of equal relevance to both urban and rural areas. Social class background has a considerable impact in many areas of life, including educational achievements, health, housing, crime and economic status. Furthermore, social class is relatively stable over time and constitutes a key factor in the inter-generational transmission of economic, cultural and social assets. Areas with a weak social class profile tend to have higher unemployment rates, are more vulnerable to the effects of economic restructuring and recession and are more likely to experience low pay, poor working conditions as well as poor housing and social environments. 

Social Class Composition is measured by five indicators: 

• the percentage of population with a primary school education only (negative association);

• the percentage of population with a third level education (positive association);

• the percentage of households headed by professionals or managerial and technical employees, including farmers with 100 acres or more (positive association);

• the percentage of households headed by semi-skilled or unskilled manual workers, including farmers with less than 30 acres (negative association);

• the mean number of persons per room (negative association).

(3) Labour market situation

Labour market situation is predominantly, but not exclusively, an urban measure. Unemployment and long-term unemployment remain the principal causes of disadvantage at national level and are responsible for the most concentrated forms of multiple disadvantage found in urban areas. In addition to the economic hardship that results from the lack of paid employment, young people living in areas with particularly high unemployment rates frequently lack positive role models. A further expression of social and economic hardship in urban unemployment blackspots is the large proportion of young families headed by a single parent. 

Labour market situation is measured by three indicators:

• the percentage of households with children aged under 15 years and headed by a single parent (negative association)

• the male unemployment rate (negative association)

• the female unemployment rate (negative association)

The fact that the HP Index derives from Census data should mean that all groups are equally and comprehensively represented in the Index. For the purposes of our analysis, a HP Index is appended to each geographic Small Area from which the address of a notification has been made. Each SA has a value for HP Index calculated as above. Patients in nursing homes, instead of attracting the deprivation index of their home address, instead are represented in the HP Index accorded to the nursing home address. For this group, we believe that their age profile may have overpowered any deprivation gradient, although this is purely from anecdotal evidence. In other words, the gradient profile may be less marked within the older age groups anyway, and the major obstacle that they face in terms of risk factors is their age.

Comment 3. Is the unit of analysis the patient/individual? Is it possible to have more than one infection for a person added to the administrative data source?

Response: Patients have unique identifiers within the dataset, and while it is possible for a patient to be infected more than once, we believe (again anecdotally), that repeat infections have been more of an issue in Ireland following relaxation of social distancing, masking and other community control measures, which came about with possibly an inflated sense of security around the degree of protection afforded by vaccination. 

Comment 4. How many hospitals are there in the study population? Is there variation in admission standards to hospital (and ICU)? If so, a random effect for hospital may need to be considered in the modeling to account for correlation in patients/outcomes in the same hospital.

Response: There are 42 acute hospitals in the Republic of Ireland. Critically, because we are a small country and the professional communities of practice are so well established through the National Clinical Programme system, there has been widespread agreement and cohesion around factors for admission to hospital and to ICU. 

Comment 5. The data collection process could be made more clear. Is there, say, a 1 page form for reporting? Are comorbidities just gleaned from that form or could comorbidities be determined from another data source? Is there data available about the type(s) of comorbidities? It is not clear how much missing and/or misclassification is present within the comorbidity variable.

Response: There is a standardised form for reporting COVID-19 in Ireland. Specific comorbidities are included on the form for inclusion as appropriate. This clarification has been added to the Methods section.

Comment 6. The outcomes are not modeled jointly, so I would not refer to this as a multivariate regression but rather multivariable regression. A multivariate analysis is not really done in this manuscript.

Response: The language has been changed to reflect this.

Comment 7. Table 1. Seem to be missing a label under health worker: ie there are 2 yes and 2 no with that variable.

Response: When a patient was a health care worker, for example it was very easy to interpret a “Y” as “Yes”, or a “N” as “No”. However, all we could tell from a missing value is that it was unlikely that the patient was a health care worker, as very clear instructions were given to data collectors to identify these individuals. In such cases, all we can do is say for definite that the binary result should be “Yes/Other”, where “Other” may mean “No” or it may mean missing data. To be honest, a “Yes” result was more significant for the purposes of follow up and occupational health notification, rather than a “No”.

Comment 8. Figure 1 and think it would be good to actually show the distribution of HP index for the study population. Further, the same kind of graphic could be done for each of the 3 waves.

Response: We felt that such a graphic may end up being too “busy”, because of smaller numbers in the five categories once we had divided them over three pandemic waves. For the purposes of our research question, we wanted to summarise the experience over the full three waves, and in order to do that, we felt that we had to combine the “Deprived” and “Very deprived” categories, in the same way as we had to combine the “Affluent” and “Very affluent” categories. We felt that this best represented what we were trying to convey.

Comment 9. The justification for not requiring an Ethics Review is not clear to me.

Response: Under the EU General Data Protection Regulations (2018), those aspects of health services research (e.g. clinical audit, health needs analysis, evaluation and monitoring etc.) which are designed for health service improvement or patient safety, and which use anonymized data from routinely-collected sources do not require clearance from an Ethics Committee. The Irish Health Services Research Regulations (2018) echo this.

Additional Comments:

Comment: Please upload a copy of Figure 4, to which you refer in your text on page 7. If the figure is no longer to be included as part of the submission please remove all reference to it within the text.

Response: Apologies. This was added in error and text has been amended. The reference should have been to Figure 3 and not Figure 4. Many thanks for spotting it.

Comment: Please remove your figures/ from within your manuscript file, leaving only the individual TIFF/EPS image files. These will be automatically included in the reviewer’s PDF

Response: Done.

Comment: Thank you for providing the following data availability statement: "The data underlying the result presented in the study are available from Joan O'Donnell (co-author), who can be contacted at joan.odonnell@hpsc.ie"

Response: In response to the comments on data availability, I believe that I am not in a position to share the data directly at this point because of General Data Protection Regulations (GDPR) which have been introduced in Europe since 2018. Because the Health Protection Surveillance Centre are the designated Data Controllers under GDPR, any request for data would need to be made through them. We have HPSC co-authors on the paper, but I gather a nominated contact would have to be someone other than a co-author, therefore I can understand that my colleague, Dr. Joan O’Donnell, would not be eligible for nomination for such a role. However, I believe that the newly-appointed director of HPSC would be an ideal contact, as he would have clinical and information governance oversight of all of the work carried out by HPSC. His name is Dr. Greg Martin and his email address is gregory.martin@hpsc.ie, or gregory.martin@hse.ie.

Many thanks for your kind attention and for your ongoing feedback. 

Formatted letter is attached separately to this submission

---

## [Decision Letter · Decision Letter 1]

10 Apr 2023

PONE-D-22-14948R1COVID-19 incidence and outcome by affluence/deprivation across three pandemic waves in Ireland: a retrospective cohort study using routinely collected dataPLOS ONE

Dear Dr. McKeown,

Thank you for submitting your manuscript to PLOS ONE. After careful consideration, we feel that it has merit but does not fully meet PLOS ONE’s publication criteria as it currently stands. Therefore, we invite you to submit a revised version of the manuscript that addresses the points raised during the review process.

ACADEMIC EDITOR: I read with interest the revised version of your manuscript. I appreciated the detailed responses you provided to the reviewers. In its current form the manuscript could be accepted, but reviewer #2 with expertise in biostatistics has raised an important issue with one of your responses. Please address this issue and resubmit your manuscript. 

We look forward to receiving your revised manuscript.

Kind regards,

Victor Daniel Miron

Academic Editor

PLOS ONE

Journal Requirements:

Reviewers' comments:

Reviewer's Responses to Questions

**Comments to the Author**

1. If the authors have adequately addressed your comments raised in a previous round of review and you feel that this manuscript is now acceptable for publication, you may indicate that here to bypass the “Comments to the Author” section, enter your conflict of interest statement in the “Confidential to Editor” section, and submit your "Accept" recommendation.

Reviewer #1: All comments have been addressed

Reviewer #2: (No Response)

2. Is the manuscript technically sound, and do the data support the conclusions?

Reviewer #1: Yes

Reviewer #2: Yes

3. Has the statistical analysis been performed appropriately and rigorously? 

Reviewer #1: Yes

Reviewer #2: Yes

4. Have the authors made all data underlying the findings in their manuscript fully available?

Reviewer #1: Yes

Reviewer #2: Yes

5. Is the manuscript presented in an intelligible fashion and written in standard English?

Reviewer #1: Yes

Reviewer #2: Yes

6. Review Comments to the Author

Reviewer #1: Thank you for the revision in response to my comments. I found all comments have been fully addressed. I have no further questions.

Reviewer #2: I was satisfied with all of the responses to my comments except one.

"Comment: I like Figure 3 very much and it seems to suggest important interactions between age and

deprivation and perhaps age, deprivation, and wave. Were interactions considered in the model? It

would seem that some should be given the exploratory data analysis.

Response: Interactions were considered in the model, particularly interaction between deprivation and age,

and that is controlled for in the model."

The authors state that they have considered interactions and that an interaction is controlled for in the model. Presumably that means that the model with the interaction is presented in the manuscript. The methods and results about the multivariable analysis do not mention interactions. Further, if the model included an interaction between deprivation and age then the aORs would be presented separately for the different combinations of age group and deprivation. That is, there should be an OR for age 40-49 HP1&HP2, age 40-40 HP4&HP5, etc. Having age group and deprivation entered as just main effects and not as an interaction means that there was no interaction in the model. As the authors state that there were interactions this statement and there methods and results presented are not in agreement and need further clarification.

7. PLOS authors have the option to publish the peer review history of their article (what does this mean?). If published, this will include your full peer review and any attached files.

Reviewer #1: No

Reviewer #2: No

---

## [Author Response · Author response to Decision Letter 1]

6 Jun 2023

Dear colleague

Very many thanks for your recent editorial advice. The only point outstanding was number 6, which I have included in full below:

6. Review Comments to the Author

Reviewer #1: Thank you for the revision in response to my comments. I found all comments have been fully addressed. I have no further questions.

Reviewer #2: I was satisfied with all of the responses to my comments except one.

"Comment: I like Figure 3 very much and it seems to suggest important interactions between age and deprivation and perhaps age, deprivation, and wave. Were interactions considered in the model? It would seem that some should be given the exploratory data analysis.

Response: Interactions were considered in the model, particularly interaction between deprivation and age, and that is controlled for in the model."

The authors state that they have considered interactions and that an interaction is controlled for in the model. Presumably that means that the model with the interaction is presented in the manuscript. The methods and results about the multivariable analysis do not mention interactions. Further, if the model included an interaction between deprivation and age then the aORs would be presented separately for the different combinations of age group and deprivation. That is, there should be an OR for age 40-49 HP1&HP2, age 40-40 HP4&HP5, etc. Having age group and deprivation entered as just main effects and not as an interaction means that there was no interaction in the model. As the authors state that there were interactions this statement and there methods and results presented are not in agreement and need further clarification.

I have had the opportunity to discuss this more closely with my colleagues and to seek additional epidemiological opinion on the issue of interaction. Because of the potential for interaction between age and deprivation, we agreed that we should include a specific interaction term including these two variable, and to determine the extent to which it may influence the overall model.

While we did observe some variable interaction, this only achieved significance for the hospital admission outcome. The overall impact on the three outcomes was substantively unchanged. In other words, age still had the greatest impact on hospital admission and mortality, and the second greatest impact on ICU admission after comorbidity, even after the model was corrected for the age*deprivation interaction.

I have amended Table 2 to include the interaction terms and have also amended the text (please see Track changes version) to draw readers’ attention to the interaction.

I hope that this helps. Very many thanks once again for the opportunity to explore this data in a more robust manner, as I believe it helps to reinforce the findings which we have observed.

Very best wishes and many thanks as ever for your ongoing interest and support.

---

## [Editor Report · Decision Letter 2]

12 Jun 2023

COVID-19 incidence and outcome by affluence/deprivation across three pandemic waves in Ireland: a retrospective cohort study using routinely collected data

PONE-D-22-14948R2

Dear Dr. McKeown,

We’re pleased to inform you that your manuscript has been judged scientifically suitable for publication and will be formally accepted for publication once it meets all outstanding technical requirements.

Kind regards,

Victor Daniel Miron

Academic Editor

PLOS ONE

---

## [Editor Report · Acceptance letter]

7 Jul 2023

PONE-D-22-14948R2 

COVID-19 incidence and outcome by affluence/deprivation across three pandemic waves in Ireland: a retrospective cohort study using routinely collected data 

Dear Dr. McKeown:

I'm pleased to inform you that your manuscript has been deemed suitable for publication in PLOS ONE. Congratulations! Your manuscript is now with our production department. 

Kind regards, 

on behalf of

Dr. Victor Daniel Miron 

Academic Editor

PLOS ONE